# The Impacts of Prenatal Mental Health Issues on Birth Outcomes during the COVID-19 Pandemic: A Scoping Review

**DOI:** 10.3390/ijerph19137670

**Published:** 2022-06-23

**Authors:** Tianqi Zhao, Hanxiao Zuo, Sandra M. Campbell, Gian S. Jhangri, Keith S. Dobson, Jessica Yijia Li, Shahirose S. Premji, Fangbiao Tao, Beibei Zhu, Shelby S. Yamamoto

**Affiliations:** 1School of Public Health, University of Alberta, Edmonton, AB T6G 1C9, Canada; hzuo3@ualberta.ca (H.Z.); scampbel@ualberta.ca (S.M.C.); gian.jhangri@ualberta.ca (G.S.J.); shelby.yamamoto@ualberta.ca (S.S.Y.); 2Department of Psychology, Faculty of Arts, University of Calgary, Calgary, AB T2N 1N4, Canada; ksdobson@ucalgary.ca (K.S.D.); yijia.li@ucalgary.ca (J.Y.L.); 3School of Nursing, Faculty of Health, York University, Toronto, ON M3J 1P3, Canada; premjis@yorku.ca; 4Department of Maternal, Child and Adolescent Health, School of Public Health, Anhui Medical University, Hefei 230038, China; taofangbiao@126.com (F.T.); a513275541@126.com (B.Z.)

**Keywords:** prenatal mental health, depression, anxiety, birth outcomes, preterm birth, low birth weight, small for gestational age, COVID-19

## Abstract

Background: The severity of the COVID-19 pandemic is likely to exacerbate mental health problems during the prenatal period and increase the risk of adverse birth outcomes. This review assessed the published literature related to the impacts of prenatal mental health issues on birth outcomes during the COVID-19 pandemic. Methods: This scoping review was conducted using PROSPERO, Cochrane Library, OVID Medline, Ovid EMBASE, OVID PsycInfo, EBSCO CINAHL, and SCOPUS. The search was conducted using controlled vocabulary and keywords representing the concepts “COVID19”, “mental health” and “birth outcomes”. The main inclusion criteria were peer-reviewed published articles from late 2019 to the end of July 2021. Results and Discussion: After removing duplicates, 642 articles were identified, of which two full texts were included for analysis. Both articles highlighted that pregnant women have experienced increasing prenatal mental health issues during the COVID-19 pandemic and, further, increased the risk of developing adverse births. This scoping review highlighted that there is a lack of research on the impact of prenatal mental health issues on birth outcomes during the pandemic. Conclusion: Given the severity of the COVID-19 pandemic and the burdens of prenatal mental health issues and adverse birth outcomes, there is an urgent need to conduct further research.

## 1. Introduction

Adverse birth outcomes, including low birth weight, preterm birth and small for gestational age, are the leading causes of infant morbidity and mortality [1]. Worldwide, 75% of neonatal mortality and morbidity is due to adverse birth outcomes [2]. Prenatal depression and anxiety increase the risk of adverse birth outcomes, such as preterm birth, low birth weight and small for gestational age [3]. According to Ding et al., women who have experienced maternal anxiety are 1.50 times more at risk of preterm birth and 1.76 times more likely to have low birth weight infants [4].

On 11 March 2020, the World Health Organization (WHO) declared the coronavirus disease 2019 (COVID-19) outbreak as a public health emergency of international concern [5]. As of 11 January 2021, the WHO reported that the cumulative infection cases worldwide have reached nearly 309 million people, with a total of more than 5 million deaths [6]. Studies have shown that infectious disease outbreaks raise the risk of developing depression and anxiety [3]. A survey conducted in Belgium among 5866 participants reported that the prevalence of depressive symptoms among pregnant women during the COVID-19 crisis was 25.3%, and this number had increased dramatically from 8.0% to 11.1% before the pandemic [7,8]. Such stressful events and conditions may also have negative impacts on birth outcomes, including preterm birth and low birth weight, and this could be due to stress-related physiological pathways during disasters [9,10].

The severity of the COVID-19 pandemic is likely to be an added burden to prenatal mental health problems and, thus, increase the risk of adverse birth outcomes. In order to address potential burdens, it is important to explore the potential impacts of prenatal mental health issues on birth outcomes during the COVID-19 pandemic. However, since the declaration of the COVID-19 pandemic, there have been limited reviews focused on prenatal mental health and its impact on adverse birth outcomes during the COVID-19 pandemic.

## 2. Materials and Methods

### 2.1. Research Aim

Purpose: This scoping review was developed to map the current literature linking prenatal mental health issues and adverse birth outcomes during the COVID-19 pandemic and to inform the relevant research and policy development.

Guiding questions: What are the impacts of prenatal mental health issues on birth outcomes during the COVID-19 pandemic?

Objective: To assess the published literature related to the impacts of prenatal mental health issues, which include symptoms of stress, anxiety and depression during pregnancy, on the adverse birth outcomes of preterm birth, low birth weight, small for gestational age, birth defects and macrosomia during the COVID-19 pandemic.

### 2.2. Methods

#### 2.2.1. Search Strategy

An expert health sciences librarian (SC) from the University of Alberta developed our search strategy and executed the final search. Seven databases were searched, including PROSPERO, Cochrane Library (CDSR and Central Register of Controlled Trials), OVID Medline, OVID EMBASE, OVID PsycInfo, EBSCO CINAHL and SCOPUS. The search was conducted using controlled vocabulary (e.g., MeSH, Emtree, etc.) and keywords representing the concepts “COVID19” and “mental health” and “birth outcomes”. The search included variations of search filters from the John W. Scott Health Sciences Library Search Filters [10,11,12,13,14]. All searches were conducted in July 2021 and adjusted appropriately for different databases. The scope was limited to articles from late 2019 to the present to capture the COVID-19 pandemic period. No other limits were applied. Our search identified a total of 894 records. All records were exported to the COVIDENCE systematic review program. We removed 252 duplicates, leaving 642 unique records. The detailed search strategies and results from each database are available in the Appendix A.

#### 2.2.2. Screening

Only peer-reviewed and published articles were included in this review. All titles and abstracts were screened for relevance by two independent reviewers (TZ and HZ). Articles that potentially discussed COVID-19 in relation to mental health, pregnant women who gave birth during the COVID-19 pandemic and birth outcomes were kept for further full-text screening. The title and abstract screening process identified 636 irrelevant records and left 6 records that met all inclusion criteria for full-text reviews.

The full-text screening was conducted by two independent reviewers (TZ and HZ). Articles that clearly discussed the targeted content were included for data extraction. Four studies were excluded at this stage, as they did not focus on our outcomes of interest, leaving two records for data extraction. Snowball searching of the final two eligible studies was also conducted, but no additional eligible studies were discovered. Figure 1 presents the PRISMA flow diagram to show our search strategy and results. Table A1 presents the PRISMA-S checklist of this scoping review.

#### 2.2.3. Data Extraction

Data extraction was performed following the Joanna Briggs Institute (JBI) recommendations to capture the data. In the data extraction spreadsheet, the information extracted included authors, year of publication, country of origin, study aims, study design, population and sample size, methods, outcomes and key findings that related to the scoping review questions.

## 3. Results

We identified two relevant articles (see Table 1). Preis et al. conducted a study in 2020 using a prospective cohort design [15]. They focused on pregnant women across the USA aged 18 years and above. Wdowiak et al. conducted a study in Poland in 2020, where longitudinal data was used, with the target population being pregnant women who received care from medical facilities in the city of Lublin [16]. Both articles found that, during the COVID-19 pandemic, there was an association between prenatal mental health disorders and adverse birth outcomes, but they approached the research from different perspectives. Whether or not such associations differed from the pre-pandemic period was not mentioned. In addition, macrosomia and birth defects were not found in this review.

Preis et al. investigated whether experiencing stress during pregnancy at the beginning of the pandemic was associated with a higher prevalence of adverse prenatal outcomes (see Table 1) [15]. This study showed that preterm birth was predicted by elevated prenatal maternal stress [15]. In this study, prenatal maternal stress was measured using the Revised Prenatal Distress Questionnaire [15]. The risk of preterm birth increased by 40% with experiences of prenatal maternal stress [15]. Overall, 1% of the variance in preterm birth and 3% in small for gestational age were attributed to elevated stress over and above other relevant maternal sociodemographic and medical characteristics [15]. Preparedness stress was found to be another factor that affected birth outcomes [15]. Preparedness stress refers to stressful feelings of not being prepared for birth or postpartum, as a result of the pandemic. It was measured using the Pandemic-Related Pregnancy Stress Scale [15]. This study shows that the risk of delivering a small gestational age infant was 66% higher among women who reported higher preparedness stress [15].

Wdowiak et al. focused on evaluating the effect of the COVID-19 pandemic on depression and whether the course of pregnancy and infant well-being would be affected by depression and the COVID-19 pandemic (see Table 1) [16]. The study used the Beck Depression Inventory (the second edition) twice during the pregnancy (first between weeks 11 and 14, and then on week 32) among the same population to assess the depressive symptoms [16]. The study showed that the severity of depression during weeks 10–13 (i.e., first trimester) and week 32 (i.e., third trimester) were significantly different from the depression severity in the second trimester, which also correlated with the COVID-19 pandemic [16]. Additionally, the severity of depression was also dependent on the interaction between the pre-pregnancy Body Mass Index (BMI) and the COVID-19 pandemic [16]. Moreover, the depression severity was negatively associated with gestational age and birth weight [16]. More severe depression and higher pre-pregnancy BMI were associated with lower gestational age at birth and lower birth weight [16]. Wdowiak et al. reported that the average birth weight and gestational age at birth were lower during the COVID-19 pandemic compared to the pre-pandemic measures: women who experienced the COVID-19 pandemic, on average, had 0.87 weeks less gestational age at delivery (*p* = 0.001), and their infants, on average, were 184.7 g lighter (*p* = 0.01), compared to infants born before the pandemic [16].

## 4. Discussion

This scoping review assessed the state of the literature on prenatal mental health disorders’ impacts on adverse birth outcomes during the COVID-19 pandemic. After reviewing the identified articles, we found that very few studies explored the association between prenatal mental health issues and adverse birth outcomes during the COVID-19 pandemic. This limited literature may be due to timing. The pandemic began early in 2020, but for prenatal mental health issues to develop and to study the impacts on adverse birth outcomes require a relatively long follow-up period. Additionally, in order to understand the unique impacts of the pandemic, longitudinal data are often needed for comparison with trends in the previous years. As the onset of the pandemic occurred relatively quickly, there might be little time to collect the longitudinal data; thus, longitudinal studies might not have been a feasible choice to assess the association.

Although limited, the published literature has demonstrated that prenatal mental health issues may have a negative impact on birth outcomes during the COVID-19 pandemic. Both articles highlighted that pregnant women experienced prenatal mental health issues, including stress, depressive symptoms and anxiety during the COVID-19 pandemic. Moreover, both articles reported that experiencing such mental health issues increased the risk of developing adverse birth outcomes, including preterm birth, low birth weight and small for gestational age [15,16].

### 4.1. Prenatal Mental Health Issues during the COVID-19 Pandemic

Pregnant women are likely to experience heightened mental health issues during any stressful events, including the COVID-19 pandemic. As the two studies reported, the average levels of prenatal maternal stress and depression have been higher during the pandemic compared to the levels before the pandemic [15,16]. Similarly, a study conducted in Poland reported that the pandemic and related restrictions exaggerated the prevalence of prenatal anxiety from 15% pre-COVID-19 pandemic to 38% during the pandemic [17]. There are three potential reasons for the exacerbation of prenatal stressors and mental health issues during the pandemic. Firstly, increasing worries and stress could be related to uncertainty regarding the effects of COVID-19 on maternal–fetal health and pregnancy. Women may feel uncertain about the effect of the virus on themselves, their fetuses and their infants [15]. Secondly, higher stress was also likely due to pandemic restrictions. Preis et al. stated that 58.0% of the participants reported canceled or altered appointments due to the pandemic, which compounded uncertainties around fetal–infant health and delivery [15]. Wdowiak et al. also hypothesized that physical distancing guidelines and social isolation would negatively impact mothers’ mental health, elevating the depressive symptoms, anxiety, insomnia and stress [16]. Furthermore, Preis et al. also reported that 40.5% of their participants experienced a pandemic-related income loss and suggested that pandemic-related financial strain was another factor that negatively affected mothers’ health [15]. The above results are consistent with other similar studies. For instance, Thayer and Gildner found that 43% of their participants reported financial stress due to the pandemic, which was significantly associated with the elevated depression scores among the pregnant women in their study [18].

### 4.2. Prenatal Mental Health Issues and Adverse Birth Outcomes

Both articles reported significant associations between prenatal mental health issues and adverse birth outcomes during the COVID-19 crisis. Preis et al. reported that pandemic-related stress was significantly associated with higher risks of preterm birth and the delivery of small for gestational age infants and also suggested that being infected with COVID-19 was a strong independent predictor of SGA infants, as infected mothers are more likely to experience higher stress [15]. Furthermore, Wdowiak et al. observed that the intensification of prenatal depressive symptoms was positively associated with low-birth-weight infants during the pandemic [16]. Such findings are consistent with previous studies, which have suggested that maternal stress exposure is related to preterm birth, as explained by the stress-related hypothesis [19,20]. However, in Preis’s study, they also found that the overall rate of preterm birth was lower than it was among the U.S. population in 2019 (7.1% and 10.2%, respectively) [15]. Preis et al. suggested it could be due to the exclusion of the youths who are under 18 years old in their sample, since younger age is one of the risk factors for preterm birth [15]. Similarly, there was some evidence also suggesting a reduction in adverse birth outcomes during the COVID-19 pandemic. For instance, the overall rate of preterm birth and SGA were lower during the COVID-19 pandemic compared to similar time periods in 2017–2019 in Botswana [21]. In Ireland, the rate of very low-birth-weight infants also reduced from 8.2 to 2.2 per 1000 live births during the COVID-19 lockdown period, compared to the periods prior to the pandemic [22]. This could also be explained by stress-related pathways. Philip et al. suggested that such reductions could contribute to “unparalleled and widespread socioenvironmental alterations to which pregnant women would have responded with appropriate behavioral and lifestyle modifications” [22]. For instance, the lockdown restrictions likely resulted in mothers resting at home and not working, in some instances. Women may also have received more partner and family support during their pregnancy compared to the pre-pandemic periods. These may have reduced their stress anxiety and depression levels of being pregnant, especially during the pandemic, and likely resulted in better infant health [22]. On the other hand, pandemic-related stress and anxiety could increase the risk of adverse birth outcomes from increased financial, family or other strains [15]. Therefore, the relationship between prenatal mental health with adverse birth outcomes during the pandemic may be a little more nuanced. Despite the association between prenatal mental health issues and adverse birth outcomes, there is a need to further investigate other potential mediators and impact factors beyond mental health issues. Furthermore, the differences in the associations between pandemic and non-pandemic periods were not mentioned in either of the included articles.

### 4.3. Limitations

This review has potential limitations. The two reviewed studies are from high-income countries. The results might not be generalizable to low- and middle-income countries or other population groups. Although Preis et al. suggested that African-American women and other women of color are at higher risk of experiencing higher pandemic-related stress and adverse birth outcomes, it remains unknown if the same trend applies to mothers from other historically disadvantaged groups during the COVID-19 pandemic [15]. Furthermore, this review did not observe any changes in the associations between prenatal mental health measures and adverse birth outcomes during the pandemic period compared to pre-pandemic periods. In addition, this review included a study with a relatively small sample size (i.e., 280 participants), so the results might not be generalizable to larger populations.

## 5. Conclusions

This scoping review highlighted that the existing knowledge on the impact of prenatal mental health issues on birth outcomes during the COVID-19 pandemic is not yet well-established. Given these early results, which show the negative effects of prenatal mental health issues on birth outcomes during the pandemic, there is an urgent need for continued research exploring and addressing prenatal mental health issues and related adverse birth outcomes during the COVID-19 pandemic.

## Figures and Tables

**Figure 1 ijerph-19-07670-f001:**
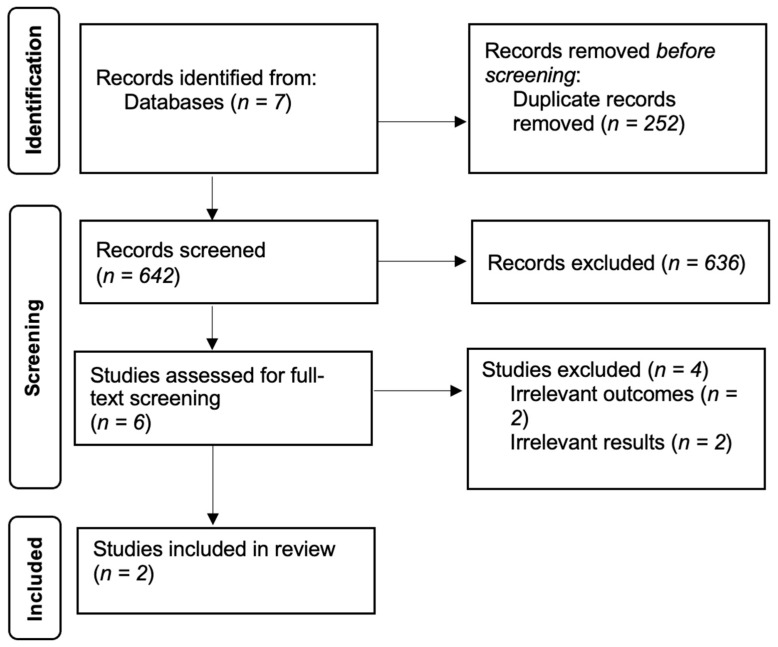
PRISMA flow diagram.

**Table 1 ijerph-19-07670-t001:** Study characteristics of the included studies.

Citation	Study Setting	Study Date	Mental Health Measure	Outcome Measure	Study Sample	Study Design	Main Results	Other Relevant Results
Preis, Mahaffey, Pati, Heiselman, & Lobel (2021) [15]	The USA	April–August, 2020	Pandemic-Related Pregnancy Stress (preparedness stress and perinatal infection stress) were measured using the Pandemic-Related Pregnancy Stress Scale; Scores calculated as mean response to items on a scale of 1 (very little) to 5 (very much) Prenatal maternal stress was assessed using the Revised Prenatal Distress Questionnaire; Scores calculated as mean response to items on a scale of 0 (not at all) to 2 (very much) General Anxiety Symptoms were assessed using the Generalized Anxiety Disorder-7; Scores calculated as the summing up scores of the responses. The cut-offs are: 0–4 minimal anxiety; 5–9 mild anxiety; 10–14 moderate anxiety; 15–21 severe anxiety	Preterm birth: gestational age at delivery less than 37 weeks. Small for gestational age: new-born weight less than 10th percentile for gestational age at delivery referring to WHO standards.	Pregnant women across the USA aged 18 years old and above N = 1367	Prospective cohort study	More than half (58.0%) of the participants reported canceled or altered appointments due to the pandemic; 40.5% experienced pandemic-related income loss; 30.3% experienced major life events during the pregnancy. Preterm birth rate among the participants: 7.1% (*n* = 97). Rate of delivering SGA infants: 8.6% (*n* = 252). Prenatal maternal stress increased the risk of preterm delivery by 40%, and explained 1% of the variance in preterm birth. Stress explained 3% of variance in SGA, and giving birth to an SGA infant was 66% greater among mothers who reported high preparedness stress.	Women who had an SGA infant are more likely to report stress than the ones who did not have an SGA infant. Outcomes were significantly related to the COVID-19 diagnosis, sociodemographic factors and medical risk factors. SGA was more than 5 times higher among women who were diagnosed with COVID-19. Anxiety symptoms, infection stress, discrimination and income loss were not related to adverse birth outcomes from the univariate analyses.
Wdowiak et al. (2021) [16]	Lublin, Poland	Longitudinal data were obtained between 1 January 2019 and 15 March 2019; 1 January 2020 and 15 March 2020	Depressive symptoms were assessed using Beck depression inventory second edition. It consists of 21 self-report questions scored from 0 to 3, the summing up score used to measure the severity of depression. The cut-offs are: 0–11 score (lack of depression); 12–19 score (mild depression); 20–25 score (severe depression	Birth outcomes were assessed using the following variables: gestational age at delivery, fetal weight and APGAR score	N = 280 (100 women with normal weight, 100 overweight women, 50 with Class 1 obesity and 30 with Class 2 obesity).	Longitudinal Study	The severity of depression was significantly higher during than before the pandemic. Women who experienced the COVID-19 pandemic on average had 0.87 week less on gestational age at delivery (*p*-value: 0.001) and their infants on average were 184.72 g lighter (*p*-value: 0.01) compared to those before the pandemic. Pre-pregnancy BMI and severity of depression in week 32 are associated with lower birth weight. The higher the pre-pregnancy BMI and the severity of depression, the lower the gestational age at delivery;	During the pandemic, pre-pregnancy BMI was positively correlated with level of depression in weeks 10–13 (r = 0.526, *p* < 0.001) and week 32 (r = 0.539, *p* < 0.001). The odds of lower APGAR (5 or 6) were significantly higher during COVID-19 pandemic than before it (OR:3.18. *p*-value: 0.009). Women with higher pre-pregnancy BMI and higher severity of depression during pregnancy have higher odds of lower APGAR.

## Data Availability

No new data were created or analyzed in this study. Data sharing was not applicable to this article.

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
