# Peer review of "The Impacts of Prenatal Mental Health Issues on Birth Outcomes during the COVID-19 Pandemic: A Scoping Review"

_ijerph, 2022, doi:10.3390/ijerph19137670_

Round 1
Reviewer 1 Report
I'm tankful for the opportunity to review the paper.
The paper is interensting and on emerging topic. The methodology is rigorous and the discussion is well written
The described future scenario is fruitful and for above points I suggest to accept the paper
Author Response
Thank you very much for your positive comments on the manuscript.
Reviewer 2 Report
This article addresses the issue of the impact of the Covid-19 pandemic on pregnancy outcomes.
The research approach is well conducted, the article clearly structured, and the results clearly presented.
However, the discussion would benefit from some improvements: the differences between the knowledge in the non-pandemic period and the results of the two articles discussed concerning the pandemic period should be highlighted more. Indeed, it would seem from what the authors report that the incidence of certain obstetrical complications was lower during the pandemic, which would require a more deep discussion of environmental factors, such as the social support that women may have received during confinement, which is potentially different from what they usually receive when the rest of the family is working or when they themselves are working late in pregnancy. Indeed, it is the differences between the usual and the pandemic period that will bring interest to this publication and not the findings of the COVID 19 period itself.
Author Response
Thank you very much for your comment. The following are our point-by-point responses:
Point 1: However, the discussion would benefit from some improvements: the differences between the knowledge in the non-pandemic period and the results of the two articles discussed concerning the pandemic period should be highlighted more.
Response 1:
The discussion has now been enhanced by the inclusion of more information related to the differences between pandemic and non-pandemic periods. We have added more information related to the differences in the rates of prenatal mental health issues, and adverse birth outcomes before and during the pandemic (lines 173-175; lines 204-208; lines 212-217). However, the differences in the association between prenatal mental health issues and adverse birth outcomes before and during the pandemic were not mentioned in the two included articles, and we have added this information in the manuscript (lines 231-233).
Point 2: Indeed, it would seem from what the authors report that the incidence of certain obstetrical complications was lower during the pandemic, which would require a more deep discussion of environmental factors, such as the social support that women may have received during confinement, which is potentially different from what they usually receive when the rest of the family is working or when they themselves are working late in pregnancy.
Response 2:
As noted by the reviewer, there was a lower prevalence of preterm birth reported in one of the studies. This could be due to the exclusion of minors who are under 18 years old from their study since younger age is also a risk factor for preterm birth. There was also some other evidence suggesting that rates of preterm birth and small for gestational age during the pandemic were lower. Further discussion of potential factors to this situation has been added in the discussion section (lines 210-212; lines 218-231).
Point 3: Indeed, it is the differences between the usual and the pandemic period that will bring interest to this publication and not the findings of the COVID 19 period itself.
Response 3:
Thank you very much for the comment and we agree that it is important to highlight the differences between non-pandemic and pandemic periods. The discussion has been edited according to the previous two points, to address the reviewer’s concern.
Reviewer 3 Report
The present review of the current literature linking prenatal mental health issues and adverse birth outcomes during the COVID-19 pandemic was written solidly and is an interesting source of information on this topic.
Principal Doubt: Surprisingly, of the entire review, only 2 publications were obtained. Were there no research papers that described the problem? To my knowledge, there are such. Maybe it's a matter of the methodology used? Please explain why this happened.
My suggestion is to include a few more fresh works on this topic: https://www.mdpi.com/1660-4601/19/9/5081. It is a rich empirical source about the described problem.
U. Nowacka, S. Kozlowski, M. Januszewski, J. Sierdzinski, A. Jakimiuk, and T. Issat, “COVID-19 Pandemic-Related Anxiety in Pregnant Women,” International Journal of Environmental Research and Public Health , vol. 18, no. 14. 2021. doi: 10.3390/ijerph18147221.
N. Rauf et al., “The Impact of the COVID-19 Pandemic on Pregnant Women with Perinatal Anxiety Symptoms in Pakistan: A Qualitative Study,” International Journal of Environmental Research and Public Health , vol. 18, no. 16. 2021. doi: 10.3390/ijerph18168237.
Besides, I have no comments to the structure or the substantive part of the work.
Author Response
Reviewer 3:
Thank you very much for your comments. The following are our point-by-point responses:
Point 1: Principal Doubt: Surprisingly, of the entire review, only 2 publications were obtained. Were there no research papers that described the problem? To my knowledge, there are such. Maybe it's a matter of the methodology used? Please explain why this happened.
My suggestion is to include a few more fresh works on this topic: https://www.mdpi.com/1660-4601/19/9/5081. It is a rich empirical source about the described problem.
Nowacka, S. Kozlowski, M. Januszewski, J. Sierdzinski, A. Jakimiuk, and T. Issat, “COVID-19 Pandemic-Related Anxiety in Pregnant Women,” International Journal of Environmental Research and Public Health , vol. 18, no. 14. 2021. doi: 10.3390/ijerph18147221.
Rauf et al., “The Impact of the COVID-19 Pandemic on Pregnant Women with Perinatal Anxiety Symptoms in Pakistan: A Qualitative Study,” International Journal of Environmental Research and Public Health , vol. 18, no. 16. 2021. doi: 10.3390/ijerph18168237.
Response 1:
We can confirm that using our search and selection processes there were only two papers that focused specifically on this topic. There could be a few reasons for including only two articles. First, the review examined literature focusing specifically on the impacts of prenatal mental health issues on adverse birth outcomes during the COVID-19 pandemic period. When we screened for the articles, there were many papers that highlighted the relationship between COVID-19 and prenatal mental health issues, but not specifically the relationship between mental health issues and adverse birth outcomes. Second, we included only peer-reviewed literature. The reason why we only focused on peer-reviewed literature was that to ensure the credibility of the work, we believed that such peer-reviewed articles would present the most reliable evidence on a newly emerging topic. It is likely that this literature will grow over time, but our paper is limited to the time frame in the review.
We do wish to thank the reviewer for bringing up the above-mentioned three articles. Two of the articles were published after we finished the search and were not included in the review (Malarkiewicz et al., 2022 & Rauf et al., 2021). We reviewed the three papers mentioned by the reviewer, and these papers were primarily focused on the impacts of the COVID-19 pandemic on mothers’ mental health, rather than the relationship between prenatal mental health issues and adverse birth outcomes during the pandemic. Although these studies were very informative, unfortunately, we had to exclude such papers as the research questions were not aligned with our inclusion criteria. This was also very common among the studies we screened, and those were not included in our paper either.
Round 2
Reviewer 2 Report
The modifications made by the authors now allow the publication of this article